

# Spatio-temporal characterisation and compensation method based on CNN and LSTM for residential travel data

Adi Alhudhaif[1] and Kemal Polat[2]

[1] Department of Computer Science, College of Computer Engineering and Sciences, Prince Sattam bin Abdulaziz University, Al-Kharj, Saudi Arabia
[2] Faculty of Engineering, Department of Electrical and Electronics Engineering, Bolu Abant Izzet Baysal University, Bolu, Turkey

## ABSTRACT

Currently, most traffic simulations require residents' travel plans as input data; however, in real scenarios, it is difficult to obtain real residents' travel behavior data for various reasons, such as a large amount of data and the protection of residents' privacy. This study proposes a method combining a convolutional neural network (CNN) and a long short-term memory network (LSTM) for analyzing and compensating spatiotemporal features in residents' travel data. By exploiting the spatial feature extraction capability of CNNs and the advantages of LSTMs in processing time-series data, the aim is to achieve a traffic simulation close to a real scenario using limited data by modeling travel time and space. The experimental results show that the method proposed in this article is closer to the real data in terms of the average traveling distance compared with the use of the modulation method and the statistical estimation method. The new strategy we propose can significantly reduce the deviation of the model from the original data, thereby significantly reducing the basic error rate by about 50%.

## INTRODUCTION

With the acceleration of urbanization, urban traffic management faces more challenges. Accurate traffic simulation is important for urban traffic planning and management, but the acquisition of actual residents' traveling data is limited by large data volume and privacy protection. Compensation for traffic simulation data is crucial in traffic research and application. For example, (1) improving data quality: In actual transportation systems, data may be lost or abnormal due to various reasons (such as sensor failures, communication interruptions, *etc.*). Through traffic simulation, these missing or abnormal data can be compensated for, thereby improving the integrity and accuracy of the data. (2) Enhanced decision support: Traffic simulation can provide a large amount of data on traffic flow, vehicle behavior, road conditions, and other aspects. These data are crucial for transportation planning, design, and management decisions. By analyzing simulation data, decision-makers can more accurately understand the operational status and potential problems of the transportation system, thereby formulating more effective transportation

Corresponding author
Adi Alhudhaif,
a.alhudhaif@psau.edu.sa

strategies. (3) Optimizing transportation system performance: By compensating and analyzing simulation data, bottlenecks and shortcomings in the transportation system can be identified. This helps to optimize and improve the transportation system, improve road traffic efficiency, reduce congestion and delay, *etc.* (4) Support for evaluation of new technologies and strategies: Traffic simulation can simulate the application effects of various new technologies and strategies (such as autonomous driving, intelligent traffic signal control, *etc.*) in transportation systems. By analyzing simulation data, the potential impact and benefits of these new technologies and strategies can be evaluated, providing decision support for practical applications. (5) Reducing experimental costs: Conducting experiments in actual transportation systems is often costly and risky. Traffic simulation can simulate various traffic scenarios and conditions in a virtual environment, and obtain a large amount of experimental data at a lower cost. By compensating and analyzing these data, valuable references and guidance can be provided for practical experiments.

Currently, traditional traffic data analysis methods mainly rely on statistical principles, and many scholars use residents' travel data as the basic data for simulation when creating traffic simulations. *Kickhofer et al. (2016)* created an open traffic simulation scenario by randomly selecting users in a survey area, analyzing the characteristics of residents' travel behavior in the area, and generating travel behaviors for the simulated users according to the probability. *Anda, Ordonez Medina & Fourie (2018)* used OD matrices from mobile phone data to simulate smart body-based urban traffic. *Guo et al. (2019)* used an online survey to collect information on residents' daily travels and used the survey data to generate travel data close to the real population in the simulation area. *Rasca (2022)* designed a questionnaire containing 36 questions to analyze the residents' age, job, commuting habits, and working characteristics, and numerical estimation of prototype users' preferences for different travel mode choices based on existing travel behavior data. *Agarwal, Ziemke & Nagel (2020)* proposed a methodology for constructing a travel diary under heterogeneous traffic conditions using intermediate blocks of hourly classified traffic counts to organize the travel demand into simulated in-scope travel demand and external travel demand, simulated in-scope travel demand directly from the travel plan survey, and external travel demand is further classified into through-travel and commuter travel, and based on this data expansion for travel time and travel location. *Wang et al. (2021)* developed a data-driven method for estimating traffic flow in large road networks by combining license plate recognition (LPR) data and taxi GPS trajectory data. *Song & Chung (2023)* proposed a unit-based traffic simulator UNIQ-SALT and an RTDG model as an adjustment model, which continuously calibrates actual traffic data and simulation results to simulate real traffic situations more finely. *Imawan et al. (2015)* proposed a MapReduce-based method to extract timeline information from road traffic data. By implementing a distributed processing strategy, scalability issues caused by data volume and complexity have been resolved.

In summary, common methods of generating data on residents' traveling include the following: (1) Face-to-face survey method. The face-to-face survey method directly collects residents' travel data, uses questionnaires and other methods to understand the characteristics and behavior of residents' travel, and then uses statistical methods to extend

the sample data to the whole population. However, since the survey data is often lower than 0.03% of the real data, the simulation results have a large error with the real scenario. (2) Statistical estimation method. Statistical extrapolation is a method of predicting the future travel situation through historical data, which can generate travel data by analyzing the historical data to predict future travel behavior. However, it is difficult and less accurate to estimate individual travel behavior from historical data of group travel. (3) GPS data mining method: GPS data mining method is a method to deduce the travel behavior of travelers by analyzing GPS track data, which can generate travel data by using the information of speed, direction, and location in GPS track data to judge the travel mode and travel purpose of travelers. However, considering user privacy issues, the method blurs the OD data through aggregation and makes data cleaning difficult due to the large amount of data and blurred data. These methods perform well on small-scale or low-dimensional data, but their efficiency and accuracy are limited when dealing with complex urban transport data.

Deep learning technology has made remarkable progress in recent years and has been widely used in fields such as image recognition and natural language processing. By simulating the way the human brain processes information, it effectively handles large-scale and complex data sets. Deep learning shows excellent ability in feature extraction, pattern recognition, *etc.*, and provides a new perspective for traffic simulation.

Convolutional neural network (CNN) has certain advantages in processing spatiotemporal data analysis, especially suitable for spatiotemporal feature analysis and compensation of resident travel data. Here are some advantages of using CNN for this analysis: (1) Extracting spatial features: CNN can effectively learn and extract spatial features from data. In residential travel data, spatial information is crucial, as factors such as traffic conditions and population density in different regions can affect travel patterns. CNN can recognize and extract these spatial features through convolution operations. (2) Capture temporal information: CNN can process temporal data and capture temporal patterns in the data. In resident travel data, time information is an important dimension, such as traffic flow and travel purposes at different time periods. CNN can capture and learn these temporal patterns through pooling operations and multi-layer structures. (3) Adapting to complex relationships: CNN can learn complex nonlinear relationships. In residential travel data, there are complex correlations between different factors, such as weather, holidays, transportation facilities, *etc.*, which can all affect travel behavior. CNN has sufficient flexibility and expressive power to learn and model these complex relationships. (4) Processing multi-scale data: CNN has a multi-level structure and can handle data of different scales. In resident travel data, there may be spatial and temporal information of different granularities, such as city level, regional level, hour level, minute level, *etc.* CNN can process data of different scales through convolutional kernels of different sizes and pooling operations. (5) Automatic feature learning: CNN can automatically learn features, reducing the need for manual feature engineering. In residential travel data, there may be a large number of features that are difficult and time-consuming to manually extract. CNN can automatically learn the optimal feature representation through a backpropagation algorithm.

In this regard, this article proposes a spatiotemporal feature analysis and compensation method for residential traveling data with CNN and LSTM, aiming to improve the authenticity and efficiency of the data. In this article, we use the spatial feature extraction capability of CNN and the advantage of LSTM in processing time series data by modeling travel time and space, aiming to achieve traffic simulation close to the real scene with limited data. The experimental results show that the algorithm proposed in this article is closer to the real data regarding average traveling distance than the traditional methods.

## RELATED WORK

Deep learning techniques have been widely used in the field of transportation. *Fan et al. (2023)* implemented cooperative control of traffic signals through deep reinforcement learning methods to improve regional traffic efficiency, and reduce vehicle time loss and waiting time. The authors proposed an index to evaluate regional traffic capacity and designed a cooperative control algorithm for traffic signals based on deep reinforcement learning to achieve this goal. Through simulation results, the authors found that the proposed DQNCC algorithm improved traffic efficiency in a specified area while reducing vehicle time loss and waiting time. *Jain, Dhingra & Joshi (2022)* discussed the problem of missing data analysis in traffic monitoring systems and the processing methods for missing data. Various missing data processing methods, such as probabilistic principal component analysis (PPCA), multi-view learning method (MVLM), and migration learning based on probabilistic principal component analysis (PPCA), and empirical analyses and method comparisons were presented for the characteristics of traffic data. In addition, some machine learning and deep learning models related to traffic data and practical applications of processing missing data in traffic management systems are introduced. *Ji et al. (2020)* describes a deep learning-based approach to predict airport network traffic in 5G scenarios. The researcher used an LSTM neural network to train on the collected 4G cellular traffic data, and the model was used to predict the network traffic. The experimental results show that the method significantly improves accuracy and cost compared to traditional methods. *Sun et al. (2020)* proposed a deep learning-based probability density function model for estimating vehicle speed and traffic information. *Cervellera, Macció & Rebora (2021)* investigated the application of deep learning and low-discrepancy sampling in agent modeling and a specific application in urban traffic simulation. The article explores the performance of deep learning structures in agent modeling applications, the evaluation of low-discrepancy sampling designs in conjunction with deep networks, and the successful application of agent models based on these two elements to high-dimensional urban traffic simulation. *Zeng (2021)* describes an intelligent traffic control system using auxiliary sensors and deep learning, which is validated by simulation experiments. The study aims to improve traffic flow efficiency by monitoring vehicles' speed through an intersection to adjust the duration of traffic signals, thus reducing the total passing time. The article details the use of deep learning algorithms for vehicle detection and tracking and auxiliary sensors for vehicle speed measurement and counting. The system's effectiveness was verified through simulation experiments. *Fu et al. (2021)* presented a deep learning-based traffic

flow rate prediction method. The authors proposed a deep learning framework (RSNet) to obtain features of road structure through CNN, temporal correlation through LSTM, and combine spatio-temporal features through MLP for prediction. The experimental results show that the method achieves better results in prediction tasks compared to traditional statistical and shallow machine learning methods. *Haydari & Yılmaz (2022)* investigated deep reinforcement learning-based traffic control applications. *Selvamanju & Shalini (2022)* investigated a deep learning-based mobile traffic prediction model (DLMTFP) in 5G networks, which aims to differentiate between a user's mobile traffic, application usage, and traffic patterns. The article proposes using a bidirectional long short-term memory (BiLSTM) model designed to predict mobile traffic and adjust the hyperparameters in the BiLSTM model to improve the prediction performance. The prediction performance of the proposed DLMTFP technique is verified through detailed simulation analysis. *Kashihara (2017)* applied augmented or deep Q-learning algorithms for traffic simulation studies. *Zhang et al. (2018)* proposed a deep learning based approach to capture the different cells by treating the traffic data as an image and using densely connected convolutional neural networks to capture the spatial and temporal dependencies between them. In addition, the article proposed a parameter matrix-based fusion scheme for learning the degree of influence of spatial and temporal dependencies. Experimental results show that the method has significantly better prediction performance regarding root mean square error than existing algorithms. *Nallaperuma et al. (2019)* introduced a Big Data-driven Smart Traffic Management Platform (STMP) based on a combination of unsupervised online incremental machine learning, deep learning, and deep reinforcement learning techniques for dealing with dynamic big data streams in the traffic domain, high-frequency unlabelled data generation, and instability of traffic conditions. *Zou & Chung (2023)* used clustering and deep learning approaches to achieve lane-level traffic flow prediction on a network-wide scale. *Trinh, Tran & Do (2022)* investigated the use of distributed deep learning to construct univariate and multivariate time series models, including LSTM, TCN, Seq2Seq, NBeats, ARIMA, and Prophet, to solve the traffic flow prediction problem. The models were implemented, and their performance was evaluated on an Irish traffic flow dataset.

In summary, applying deep learning techniques in the transport field has shown a wide range of prospects. The research covers many aspects, such as traffic signal control, missing data processing, network traffic prediction, vehicle speed estimation, agent modeling, intelligent traffic control systems, traffic flow prediction, mobile traffic prediction in 5G networks, and big data-driven intelligent traffic management platforms. These studies have improved traffic efficiency and reduced waiting time but also processed and analyzed traffic data through different deep learning models and methods, providing more accurate information and predictions for traffic management and planning.

## PRELIMINARY

### Preparation of data

In preparing the data, we need to collect two main types of data: time series data and spatial data. Examples of data are shown in Table 1. Time series data include traffic flow, vehicle

| Table 1 Preparatory data. | | |
| --- | --- | --- |
| Data type | Example content | Data format |
| Time series data | Traffic flow and speed | CSV, Excel |
| Spatial data | Road network and traffic lights | Shapefile, GeoJSON, Image data |

speed, traveling time, *etc.* These data are usually recorded by time stamp (date and time) and stored in CSV or Excel format. Spatial data, on the other hand, covers information such as road network structure, traffic signal locations, and traffic hotspots, which are usually stored in GIS data formats such as Shapefile or GeoJSON, or as image data (*e.g.*, traffic heat maps).

The input data used by CNN: The CNN part is mainly used to process spatial features, so the input data includes satellite images of road networks, traffic heat maps, or graphical representations of traffic flow distribution. These data can help the model understand the distribution patterns of traffic flow in different geographical regions, such as the congestion situation of major traffic roads and intersections in cities.

The input data used by LSTM: The LSTM part is used to process time series data, and the input data mainly includes traffic information about time changes, such as historical traffic flow, speed, accident reports, *etc.* This part of the data enables the model to capture the dynamic changes in traffic flow over time, including seasonal changes, daily peak hour changes, and so on.

Through this approach, the model can simultaneously consider the spatial and temporal dimensions of traffic data, thereby improving the accuracy and efficiency of traffic state prediction.

In our study, the image data used mainly came from urban traffic monitoring cameras, and the resolution of these images is usually $640 \times 480$ pixels. For traffic flow and other related time series data, we collect data every 5 min, covering a time range of several consecutive months. All data has undergone appropriate preprocessing steps, including standardization, before being input into the model to ensure the effectiveness and accuracy of model training. We promise to provide more detailed data specifications and descriptions of preprocessing methods in future work, so that readers and other researchers can better understand and reproduce our research results.

## Data pre-processing

We carried out a meticulous data preprocessing exercise to ensure that CNN and long short-term memory networks (LSTM) could effectively process the data. First, we removed all irrelevant and duplicate records from the dataset. For missing values, we carefully analyzed the data according to their characteristics and importance and selected the most suitable filling method or directly deleted the records containing missing values. In addition, we unified the format of all the data to ensure that the CNN could process the spatial data, and at the same time, we ensured that the time-series data conformed to the input format of the LSTM. To reduce the possible effects of different magnitudes of data, we standardized or normalized all numerical data. Finally, we identified and dealt with

outliers through data visualization techniques to improve data quality, thus laying a solid foundation for the next model training. Algorithm 1 describes the specific algorithm steps.

---

**Algorithm 1** Data Preprocessing

---

1: Initialize $D'$ as an empty dataset
2: **for** each record $d$ in $D$ **do**
3:     **if** $d$ is a duplicate **then**
4:         Remove $d$ from $D$
5:     **else**
6:         **for** each feature $f$ in $d$ **do**
7:             **if** $f$ is missing **then**
8:                 **if** $f$ is an important feature **then**
9:                     Fill $f$ using interpolation methods
10:                 **else**
11:                     Remove $d$ from $D$
12:                     Go to the next record
13:                 **end if**
14:             **end if**
15:         **end for**
16:         **if** $d$ is valid **then**
17:             $d$ to match CNN and LSTM input formats $t_f$ and $s_f$
18:             Normalize numerical features in $d$
19:             Handle anomalies in $d$ using visualisation techniques
20:             Add processed $d$ to $D'$
21:         **end if**
22:     **end if**
23: **end for**
24: **return** $D'$

---

## METHODOLOGY

In this study, a traffic data analysis model combining a CNN and a LSTM is proposed. The model aims to take advantage of the powerful spatial feature extraction capability of CNNs and the strength of LSTMs in time series analysis to more accurately predict and analyse analyze urban traffic flow patterns.

Figure 1 depicts the architecture of the deep learning model constructed in this study, which incorporates a CNN and a LSTM designed to perform comprehensive analysis of urban traffic data. In the first stage of the model, the input traffic image data is preprocessed to fit the input requirements of the CNN and passed through multiple convolutional layers containing filters of different sizes to capture spatial patterns ranging from coarse to fine-grained. Each convolutional layer is followed by a pooling layer to reduce the feature

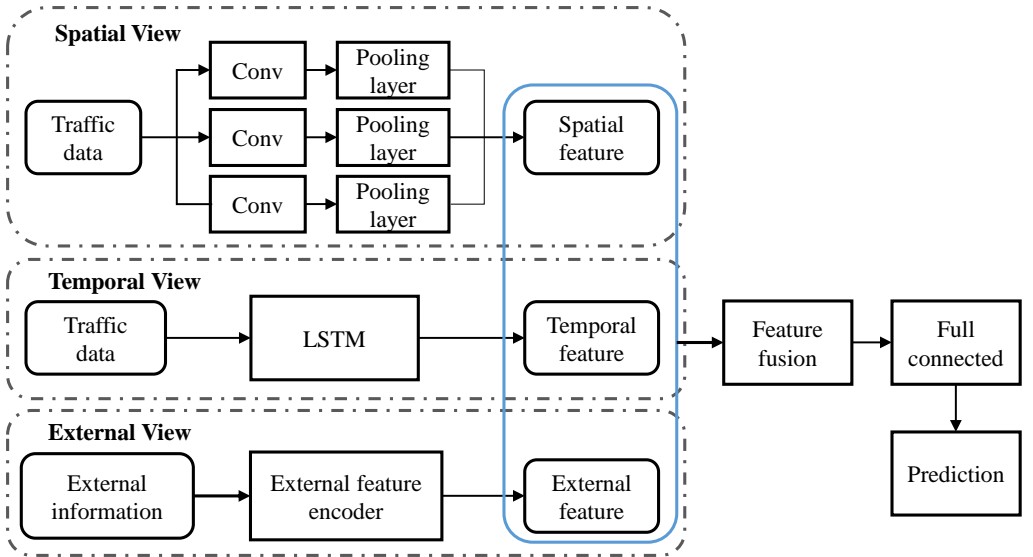

**Figure 1** Model architecture: (A) Beijing traffic heat map (B) Beijing traffic flow map.

dimensions and enhance the invariance properties of the model. After layers of convolution and pooling, the resulting spatial feature map significantly reduces the dimensionality of the data while maintaining key spatial information.

Next, these CNN-processed spatial features are fed into the LSTM network. At this stage, the model focuses on analyzing the features over time, using the recursive structure of the LSTM network to capture the dynamic dependencies of time-series data, such as traffic flow and speed. Due to the special design of LSTM, it can effectively deal with long-term temporal dependencies, thus capturing deeper temporal dynamic features. Spatial and temporal features are integrated into a fusion layer, which uses specific fusion strategies such as weighted sums or concatenation to combine information from the CNN and LSTM. The fused features are then passed through a series of fully connected layers capable of learning non-linear combinations between features for higher levels of abstraction. In the final stage of the model, an output layer is responsible for converting the processed data into a traffic pattern prediction that can assist in traffic management decisions. The model demonstrates state-of-the-art performance in traffic data analysis through an exhaustive training and validation process, providing an innovative solution for real-time traffic state prediction and traffic congestion problems.

## Extracting spatial features using CNN

CNNs are widely used in the traffic domain due to their ability to extract spatial features from image data. CNNs can automatically learn the hierarchical features of a traffic scene, from simple edges to complex road and vehicle patterns, through their convolutional layers. This feature allows CNNs to exhibit strong generalization capabilities when processing traffic images and to identify key traffic features in different environments. Due to the complexity of traffic data, traditional feature engineering is time-consuming and often

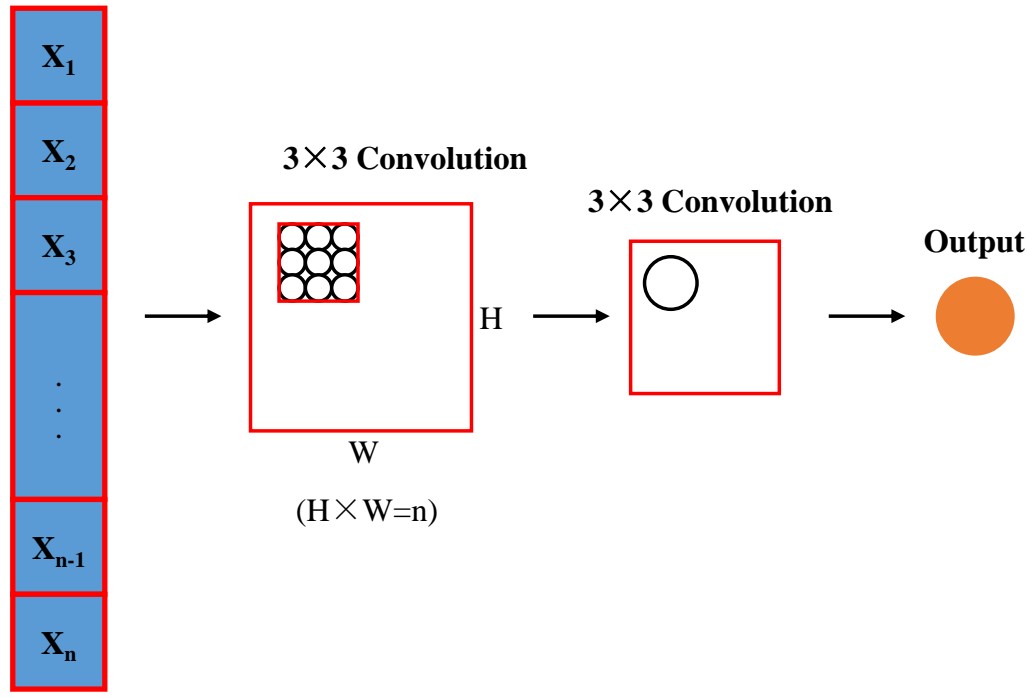

**Figure 2** Convolutional layer architecture.

relies on expert knowledge. CNN's automatic feature extraction mechanism overcomes this limitation and provides an efficient solution for in-depth analysis of traffic states. Therefore, given its superior feature extraction capability and automation characteristics, this study selects CNN as the core technology for traffic image data analysis.

## CNN architecture

This study uses a CNN model containing three convolutional layers to extract spatial features of traffic images. Each convolutional layer uses a rectified linear unit (ReLU) activation function and is connected to a maximum pooling operation. The convolutional layer architecture is shown in Fig. 2.

    **First convolutional layer configuration.** This layer employs 64 filters of size $3 \times 3$ with the step size set to 1 to ensure that detailed features in the traffic image, such as vehicle shapes and road edges, are captured. In addition, zero padding was used to maintain the size of the feature map. After each convolution operation, the ReLU activation function was applied to provide nonlinear processing capabilities to help the network learn more complex feature representations. Next, a $2 \times 2$ window was used for the maximum pooling operation, with the step size set to 2, to reduce the dimensionality of the feature maps and improve computational efficiency.

    **Second convolutional layer configuration.** In deeper network structure, the second convolutional layer uses 128 $5 \times 5$ filters with a step size of 1, and no zero padding is applied. This layer aims to extract higher level higher-leve features such as traffic density and traffic

flow direction. Again, the ReLU activation function is used to introduce nonlinearities and maximum pooling of the same size and step size is performed.

**Third convolutional layer configuration.** The last convolutional layer is configured with 256 7 × 7 filters with a step size of 1 and no zero padding, with the aim of extracting to extrac global features in the image, such as the overall traffic scene layout. The ReLU activation function and the 2 × 2 max-pooling operation are again used to extract the key features and reduce the feature dimensions.

The configuration of these layers is chosen to enable layer-by-layer feature extraction from the concrete to the abstract, with each layer designed to balance the model's representational capabilities with computational efficiency. The use of the ReLU activation function reduces the problem of gradient vanishing during the training process, while both the increasing number of filters and the decreasing pooling operation help the model to learn learning more complex traffic patterns while retaining key spatial information.

**Choice of activation function.** To enhance the nonlinear capability of the model in extracting spatial features in traffic images, this study employs a modified linear unit (ReLU) activation function after each convolutional layer. The ReLU function, by its definition $f(x) = max(0,x)$, can effectively set the negative part of the input signal to zero while keeping the positive part unchanged. This property not only reduces the computational complexity but also maintains a large gradient during the backpropagation process, avoiding the problem of gradient vanishing that is common in deep networks. In addition, the linear property of ReLU in the positive interval accelerates the convergence speed and makes the model training more efficient. In the application of traffic image analysis, the complexity of the traffic scene requires that the model must quickly adapt to a large amount of high-dimensional data. Therefore, the simplicity and efficiency of ReLU make it an ideal choice for improving the performance of traffic data analysis models.

**Pooling operation.** In this study, to effectively reduce the number of parameters in the model and enhance its robustness to spatial variability, we employ a maximum pooling operation of 2 × 2 size with a step size of 2 after each convolutional layer. The maximum pooling method selects the maximum value of each 2 × 2 window in the feature map, which is particularly suitable for emphasizing salient features in traffic images, such as vehicle concentration areas and traffic congestion points. In addition, we chose the strategy of not using padding (Valid Padding) to reduce the influence of the edges of the feature maps, thus making the model more focused on the essential features in the central region of the traffic image. This decision helps the model to focus on the core traffic dynamics rather than the edge parts of the image when analyzing urban traffic flow patterns, thus improving the model's prediction accuracy and generalization ability in complex urban traffic environments.

**Application of LSTM for processing time series.** In order to parse the time series properties in traffic data, this study introduces the LSTM. The LSTM can learn the long-term dependencies in time series through its structure's forgetting gates, input gates, and output gates, thus effectively capturing the dynamics of traffic flow, speed, and other traffic metrics over time. Specifically, the update of an LSTM cell can be formulated as follows:

$$f_t = \sigma \left( W_f \cdot \left[ h_{t-1}, x_t \right] + b_f \right) \tag{1}$$

$$i_t = \sigma \left( W_i \cdot \left[ h_{t-1}, x_t \right] + b_i \right) \tag{2}$$

$$\tilde{C}_t = \tanh \left( W_C \cdot \left[ h_{t-1}, x_t \right] + b_C \right) \tag{3}$$

$$C_t = f_t * C_{t-1} + i_t * \tilde{C}_t \tag{4}$$

$$o_t = \sigma \left( W_o \cdot \left[ h_{t-1}, x_t \right] + b_o \right) \tag{5}$$

$$h_t = o_t * \tanh \left( C_t \right) \tag{6}$$

where, $f_t, i_t, o_t$ is the forget gate, input gate and output gate respectively, $\sigma$ is the sigmoid function, $\tilde{C}_t, C_t$ is the cell state and $h_t$ is the hidden state.

LSTM was chosen over other models, such as traditional RNN, because it is more effective in handling long-term sequence data and can better avoid the gradient vanishing problem. This is crucial for this study because time-series features of traffic data often contain complex time-dependent relationships that are critical for accurately predicting traffic flow and status.

## LSTM processing of time series

This study uses a LSTM network to process the time-series features of traffic data. The LSTM efficiently manages the long-term storage and short-term memory of information through its unique gating structure, including forgetting gates, input gates, and output gates. At each time step, the LSTM cell decides which information is retained or forgotten and how new information is integrated into the cell state. This enables the LSTM to capture complex time dependencies such as fluctuations in traffic flow and speed over time, thus providing strong support for traffic flow prediction and analysis. The LSTM architecture is shown in Fig. 3.

As shown in Fig. 3, the LSTM network architecture used in this article consists of multiple layers, each containing several LSTM units. Each cell consists of three main parts: a forget gate (responsible for deciding whether to retain or discard information in the cell state), an input gate (controlling the amount of new information added), and an output gate (determining the output based on the current cell state and the previous hidden state). In addition, each LSTM cell maintains an internal cell state passed through the time series, effectively capturing long-term dependencies. The architecture is particularly well suited

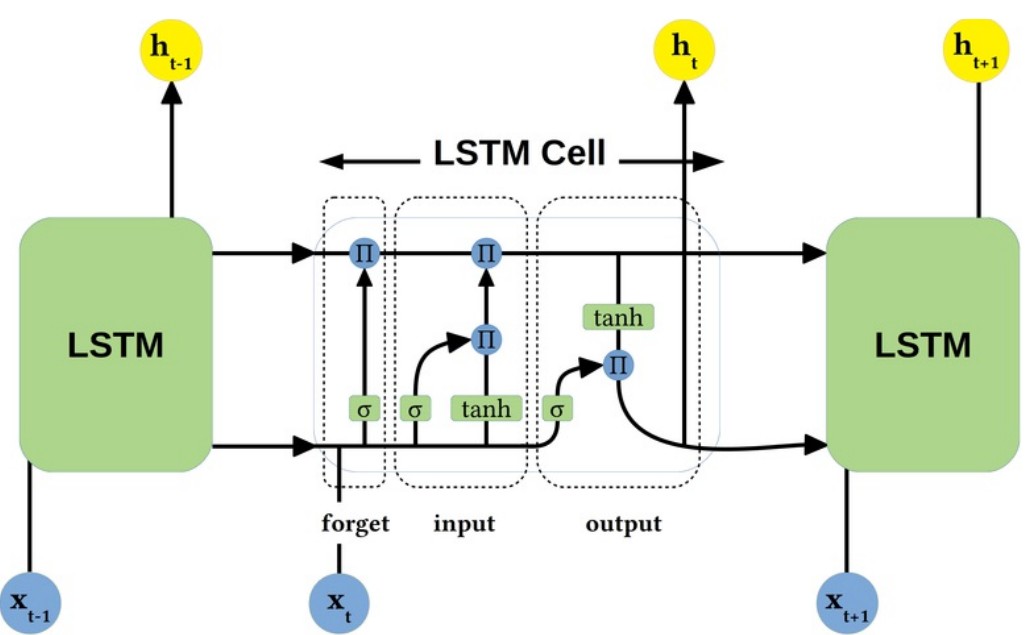

**Figure 3  LSTM architecture.**

to deal with time series in traffic data, such as changes in traffic volume and speed over time, enabling the model to predict future traffic conditions accurately.

**Oblivion gate.** In this study, the forgetting gate of the LSTM is responsible for deciding which information should be removed from the cell state at each time step. The following equation can express the operation of the oblivion gate:

$$f_t = \sigma \left( W_f \cdot \left[ h_{t-1}, x_t \right] + b_f \right) \tag{7}$$

where, $f_t$ represents the output of the oblivion gate, $\sigma$ is the sigmoid function to ensure the output value is between 0 and 1. $W_f$ and $b_f$ are the weights and biases of the forgetting gate, respectively, $h_{t-1}$ represents the hidden state of the previous time step, and $x_t$ represents the input of the current time step. The output of this gate determines which information from the previous state is retained and which is forgotten. This mechanism allows the LSTM to dynamically adjust its memory to selectively retain information critical for predicting future traffic flows and trends in response to the needs of traffic data analysis.

**Input gate.** The input gate of the LSTM in this study is responsible for deciding which new information will be added to the cell state. The configuration of the input gate consists of a weight matrix, bias terms, and a sigmoid activation function that ensures that the output value is between 0 and 1. Specifically, the operation of the input gate is described by the following equation:

$$i_t = \sigma \left( W_i \cdot \left[ h_{t-1}, x_t \right] + b_i \right) \tag{8}$$

$$\tilde{C}_t = \tanh \left( W_C \cdot \left[ h_{t-1}, x_t \right] + b_C \right) \tag{9}$$

$$C_t = f_t * C_{t-1} + i_t * \tilde{C}_t \tag{10}$$

where $i_t$ denotes the input to the input gate, $\tilde{C}_t$ denotes the candidate cell state, and $C_t$ denotes the updated cell state. This mechanism allows the model to dynamically adjust its internal state based on current inputs and past information, providing accurate control for this study's time series analysis of traffic data.

**Output gate.** In the LSTM model of this study, the output gate is responsible for determining the final output for each time step based on the cell state. The configuration of the output gate consists of a weight matrix, bias terms, and a sigmoid activation function, as well as a tanh function to handle the cell states. The details are as follows:

$$o_t = \sigma \left( W_o \cdot [h_{t-1}, x_t] + b_o \right) \tag{11}$$

$$h_t = o_t * \tanh(C_t) \tag{12}$$

where $o_t$ denotes the activation value of the output gate and $h_t$ denotes the output of the LSTM unit. The role of the output gate is to filter out the features that are important for the current time step and the model can make accurate traffic flow predictions based on past and current information.

### Modelling adjustments

The model-tuning process involves parameter selection and configuration optimization. For the CNN part, we determined the number and size of filters in the convolutional layer by grid search and optimized the learning rate and regularisation parameters. For the LSTM part, we adjusted the number of hidden layer units and the learning rate. A cross-entropy loss function was used for model training, defined as:

$$L = -\sum \left( y \log(\hat{y}) + (1-y) \log(1-\hat{y}) \right) \tag{13}$$

where $y$ denotes the real labels and $\hat{y}$ denotes the model prediction. This article chooses Adam as the optimizer whose adaptive learning rate can speed up convergence and improve model performance. In addition, an early stopping strategy is employed to avoid overfitting. The selection of these tuning strategies is based on the understanding of traffic data characteristics and the results of pre-experimentation, aiming to improve the model's accuracy and generalization on the traffic flow prediction task.

### Forecasting and compensation

The core objective of this article is to accurately predict and effectively compensate traffic data through deep learning models. The data preprocessing involves collecting many traffic images and time-series data such as traffic volume, speed, and accident rate. Models combining CNN and LSTM extract spatial and temporal features of these data. During prediction, these features are integrated and processed through a fully connected layer to generate predictions of traffic flow patterns.

For data compensation, we use the following formula:

$$\hat{X}_t = LSTM(X_{t-1}, X_{t-2}, \ldots, X_{t-n}) \tag{14}$$

where $\hat{X}_t$ denotes the predicted value at the point in time is used to compensate for the indeed data. This compensation mechanism relies on the ability of the LSTM to infer missing values based on known data, maintaining the continuity and accuracy of traffic flow predictions. This approach is particularly applicable in sensor failure or data loss cases, ensuring the integrity of the analysis and prediction process.

## Combining CNNs and LSTMs

This article uses a combination of CNNs and LSTMs to analyse analyze and predict traffic flow patterns. The CNNs first process traffic image data to extract key spatial features such as road congestion and traffic flow patterns. These features are then fed into the LSTM and other time-series data (*e.g.*, vehicle speeds and flow rates). The LSTM analyses the trends of these features over time, taking advantage of its ability to capture long-term dependencies. Finally, the features extracted by the CNN and LSTM are fused and a fully connected layer is used to generate the final traffic flow prediction. This method effectively combines the capabilities of CNN in spatial feature extraction and the advantages of LSTM in time series analysis to provide more accurate and comprehensive decision support for traffic management.

## External information coding

The weather information is discrete, so the manuscript uses mono-thermal coding to represent it. For example, we classify the weather into $N_{wea}$ types (sunny, rainy, *etc.*), so each type corresponds to a single thermocoded $O_{wea}$ of $N_{wea}$ dimension size. Holiday information is also a discrete feature, so the manuscript also uses unique hot coding to represent it. For example, we divide holidays into $N_{day}$ types (working days and non-working days), so each type corresponds to a unique hot-coded $O_{day}$ of $N_{day}$ dimension size. In order to obtain the final coding representation of external information, we splice $O_{wea}$ and $O_{d}ay$ into a fixed-length vector $Z = concat(O_{wea}, O_{day}) \in \mathbb{R}^{(N_{wea}+N_{day})}$, and then use a two-layer fully connected neural network to transform the vector into the final vector ocode, as shown in the following equation:

$$ocode = W_m^2 ReLU(W_m^1 Z + b_m^1) + b_m^2 \tag{15}$$

where $W_m^1 \in \mathbb{R}^{(d_m^1 \times (N_{wea}+N_{day}))}$ and $b_m^1 \in \mathbb{R}^{(d_m^1)}$ are parameters of the first layer neural network, $W_m^2 \in \mathbb{R}^{(d_m^2 \times d_m^1)}$ and $b_m^2 \in \mathbb{R}^{(d_m^2)}$ are parameters of the second layer neural network, and $d_m$ represents the dimension size of the neural network. In the experimental part, the revised draft uses two kinds of external contextual data information: weather and vacation.

**Table 2   Example of data structure for residential trips.**

# EXPERIMENTS

## Experimental setup

**Experimental data**. The experimental data in this study is randomly generated using the travel plan generator provided by multi-agent transport simulation (MATSim) (*Rahimi et al., 2021*) with a data size of 1.5 million entries and a simulated city area of 490 km$^2$. The data structure is shown in Table 2.

**External information data.** This experimental uses two types of external contextual information: weather and holidays. First, the weather information is extracted through the open source website (*Gearhart, 2023*). Similar to the treatment of weather (*Geng et al., 2019*), this experiment sets the number of weather types to 16, that is, $N_{wea} = 16$. Second, for holiday information, this lab sets the number of holiday types to 2, that is, $N_{day} = 2$.

**Evaluation metrics.** This experiment mainly compares the performance of different methods on three evaluation indicators, which are mean absolute error (MAE) and mean absolute percent error (MAPE), and mean absolute relative error (MARE). Specifically, assuming that the real value is $y = \{y^i\}$ and the estimated value is $\hat{y} = \hat{y}^i$, then the calculation formulas of these three metrics are as follows: $MAE(y, \hat{y}) = \frac{1}{N}\sum_{i-1}^{N}|y^i - \hat{y}^i|$, $MAPE(y, \hat{y}) = \frac{1}{N}\sum_{i-1}^{N}\left|\frac{y^i - \hat{y}^i}{y^i}\right|$, $MARE(y, \hat{y}) = \frac{1}{N}\frac{\sum_{i=1}^{N}|y^i - \hat{y}^i|}{\sum_{i=1}^{N}|y^i|}$. In this study, MAE is used as the loss function of training STSim model. In addition, the selected parameter learning optimization algorithm is Adam, and the selected batch size is 1024. Finally, the learning rate of the model parameters is initialized to 0.01, and the learning rate is reduced by 0.2 every two rounds.

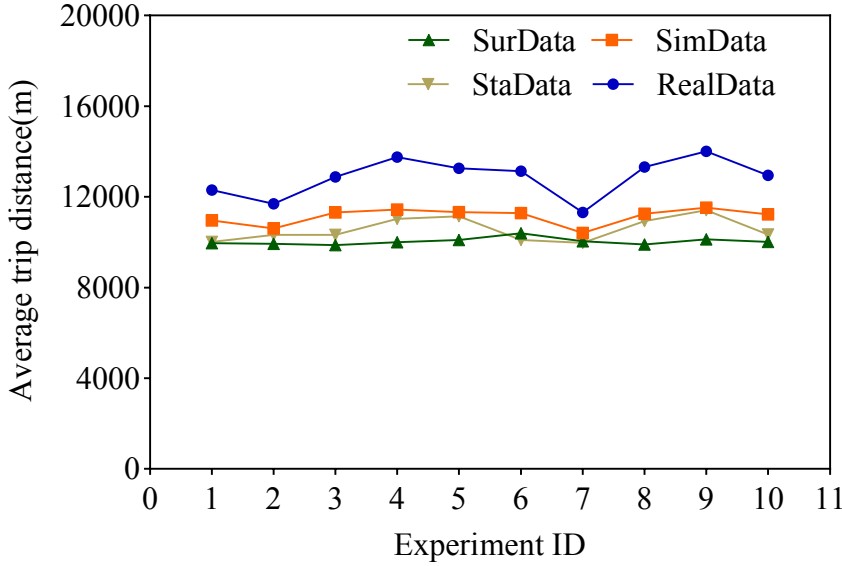

**Figure 4** Comparison of average distance travelled.

**Experimental environment.** The experiments in this study are conducted using Python programming language with version 3.10.0. All the experiments in this paper are executed on Intel(R) Core(TM) i7-12700HQ CPU @ 2.80 GHz and 16.0GB RAM. The experimental equipment is a Dell laptop with Windows 10 64-bit operating system.

## Experimental results and analyses

In this research article, we highlight the idea of using the average voyage of service providers as one of the key elements in assessing accuracy. Subsequently, we conducted a comparative study between the questionnaire methodology and our innovative proposal in this article under a similar setting. For the purpose of this study, we successfully generated 10 simulated datasets SimData based on CNN and LSTM, as well as datasets SurData originating from the traditional questionnaire survey method and StaData from the statistical estimation method. Through an in-depth discussion of the results obtained, we have come to an important conclusion: compared to the data obtained using the questionnaire survey method, the The novel algorithm we propose in this paper for the automatic generation of traffic travelling traveling data based on deep learning techniques–CNN and LSTM–presents information that is closer to the original data. As shown in Fig. 4, this difference is obvious. In addition, according to Table 3, it can be seen that this new strategy we propose significantly reduces the condition of model deviation from the original data, which in turn leads to a significant reduction in the fundamental error rate, which is as high as about 50%.

We believe that this method of using deep learning techniques to process traffic data is very promising. Traditional methods of collecting transportation data, such as questionnaire surveys or statistical estimates, are often limited by various factors such as manpower, material resources, and time. Deep learning technology can automatically

**Table 3 Error analysis.**

| Experiment ID | Average trip distance (m) | SimData error rate | SurData error rate | StaData error rate |
|---|---|---|---|---|
| 1 | 12,301 | 0.109 | 0.189 | 0.186 |
| 2 | 11,687 | 0.093 | 0.149 | 0.116 |
| 3 | 12,886 | 0.122 | 0.233 | 0.198 |
| 4 | 13,751 | 0.168 | 0.272 | 0.198 |
| 5 | 13,267 | 0.146 | 0.238 | 0.159 |
| 6 | 13,129 | 0.141 | 0.208 | 0.231 |
| 7 | 11,315 | 0.08 | 0.111 | 0.119 |
| 8 | 13,315 | 0.155 | 0.256 | 0.178 |
| 9 | 14,003 | 0.177 | 0.277 | 0.185 |
| 10 | 12,950 | 0.133 | 0.227 | 0.201 |

extract useful information from a large amount of data without the need for tedious manual processing. In addition, deep learning models such as CNN and LSTM can handle complex spatial and temporal dependencies, which traditional methods find difficult to achieve.

We believe this is because CNN excels in processing image and spatial data, while LSTM excels in processing sequence and temporal data. Combining these two can fully leverage their advantages and capture the characteristics of traffic data more comprehensively. The experimental results also prove this point, and this combined method has significant effects in reducing model bias and lowering the basic error rate. The following is a detailed comparative analysis.

(1) The traditional questionnaire survey method has a series of distinctive features and significant disadvantages in generating residents' travelling data. First, the method has directness, through direct communication with respondents, it can obtain first-hand information on residents' travel behaviour and preferences, which helps the researcher to deeply understand the respondents' real thoughts and habits. Secondly, the questionnaire survey is targeted, and the researcher can design the questionnaire content according to the specific needs of the study, so as to accurately collect specific data on the purpose, mode, time and other aspects of travelling. Finally, the broadness of questionnaire survey makes it able to cover a wider range of areas and people, so as to obtain the travelling data of residents in a wide range. However, the traditional questionnaire survey method also has some obvious disadvantages. Firstly, low response rate is a problem that cannot be ignored, as respondents may be reluctant to participate or complete the questionnaire due to factors such as time and interest, resulting in a low response rate for data collection. Secondly, the issue of data accuracy is also a concern. Respondents may provide inaccurate information due to memory bias, differences in understanding, or intentionally, thus affecting the authenticity and reliability of the data. In addition, questionnaire surveys are more time- and resource-consuming in terms of time and resources consumption. From designing the questionnaire, distributing the questionnaire to collecting and processing the data, the whole process is time-consuming and labour-intensive, which is especially obvious when conducting large-scale surveys. Meanwhile, sample bias is also a potential problem.

Non-random sample selection or low response rates may result in an under-representative sample, and the results may not accurately reflect the actual situation of the entire target population. Finally, questionnaire surveys usually focus on the collection of quantitative data, and there may be analytical limitations in digging deeper into the deeper reasons or motivations behind residents' travel behaviour. In addition, questionnaire surveys also involve privacy and sensitivity issues, and some residents may feel uneasy about providing their personal travelling information, especially when personal privacy or sensitive topics are involved, which may further affect the completion of the questionnaire and the collection of data. Therefore, when using the traditional questionnaire survey method, the researcher needs to fully consider these issues and take appropriate measures to improve the quality and reliability of the data.

(2) The statistical estimation method significantly relies on historical data. It attempts to capture the patterns and trends in the historical travel data through in-depth analysis, and then predicts future travel behaviour. The advantage of this method is that it does not require real-time or first-hand travel data, thus reducing the complexity and cost of data collection. In addition, the statistical imputation method performs well when dealing with long-term trend analysis, and is particularly suitable for analysing and predicting macro trends such as seasonal changes and annual growth in travel behaviour. However, the statistical imputation method also has some obvious drawbacks. First, the accuracy of its forecasting results is limited by the quality and quantity of historical data. If there are missing, incorrect or incomplete historical data, the accuracy of the forecast results will be seriously affected and may not accurately reflect the actual changes in the future. Secondly, the statistical projection method tends to ignore changes at the micro level. It focuses more on long-term and macro trends and may not be able to capture in time the impact of short-term contingencies or temporary changes on traffic flow. Finally, the statistical estimation method has difficulties in predicting individual behaviour. As it is mainly based on statistical analysis of historical data, it is difficult to accurately predict individual travel choices and behavioural patterns, and usually only provides macro-level trend predictions. Therefore, when using the statistical estimation method for traffic travel behaviour analysis, we need to be fully aware of its characteristics and shortcomings, and choose and apply it carefully in the light of the actual situation. At the same time, in order to improve the accuracy of prediction, we also need to continuously optimise and improve the collection and processing methods of historical data, as well as explore more accurate prediction models and algorithms.

(3) The method proposed in this paper cleverly combines CNN and LSTM. Among them, CNNs are used to process spatial features and effectively capture the spatial distribution patterns of traffic data; while LSTMs are responsible for processing time-series data and are able to capture the dynamic changes of traffic flow over time. This combination enables the method to deeply analyse traffic data and achieve more accurate predictions. In addition, the method is able to simultaneously consider the complex relationship between spatial distribution and temporal changes, and automatically extract features through the deep learning model, thus avoiding tedious manual feature engineering and improving the accuracy of the model for traffic flow prediction. In addition, the method has the ability

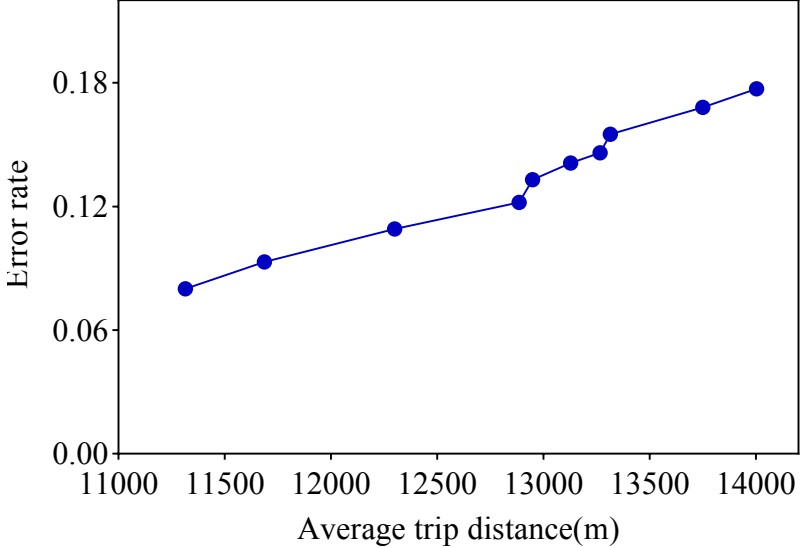

**Figure 5** Error rate *versus* average distance travelled.

to adapt dynamically to effectively deal with and compensate for missing or abnormal situations in the data, thus improving the completeness and accuracy of the data. This ability makes the method more robust in practical applications and able to cope with a variety of complex data environments. Reduces the need for large-scale data collection and significantly improves efficiency compared to survey methods. Ability to handle complex spatial and temporal dependencies with performance that is difficult to achieve with traditional methods. Compared to traditional survey methods and statistical estimation methods, the method in this paper reduces the need for large-scale data collection and significantly improves the efficiency of analyses through deep learning techniques.

After analyzing Table 3 in detail, we found that if the average distance traveled by the actual service provider is larger, the error between the distance traveled generated using CNN and LSTM and the average distance traveled by the actual service provider increases accordingly. We list the 10 tests in this study in order of proximity to the actual service provider's average tele-trip distance, and this ordering is reflected in the error rate evolution chart in Fig. 5.

Observing the visualization in Fig. 5, we can observe that the error rate of our proposed travel planning and scheduling system climbs as the average distance traveled continues to rise. One potential reason for this may be that some service providers have traveled much more than their average long-distance travel routes on a given date. To explore this issue in depth, we selected 0.001% of the samples in the generated data, extended the distance of the travel destinations of these samples, and analyzed the error rates before and after the experiment. The final results are presented in Fig. 6.

A closer look at Fig. 6 shows that the error rate after the data extension process is significantly reduced, which strongly supports our initial conjecture. In order to further validate that the method proposed in this paper is superior to the existing state-of-the-art

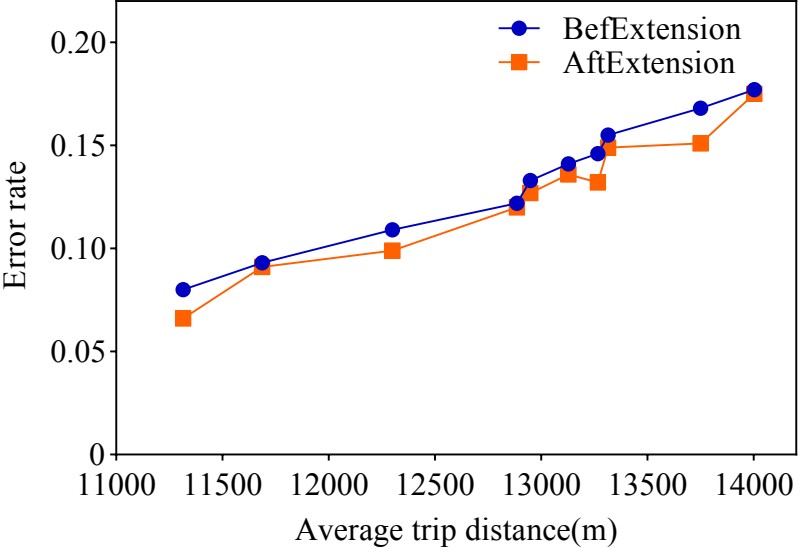

**Figure 6** Comparison of data before and after extension.

methods, this paper also conducts a comparison experiment with the literature *Song & Chung (2023)*. *Song & Chung (2023)* proposes a cell-based traffic simulator UNIQ-SALT and introduces the Recurrent Traffic Demand Generation (RTDG) model as an adjustment model for the simulation inputs.The RTDG model improves the accuracy of the simulation by continually calibrating the simulation results with the actual traffic data until it reaches the preset target error rate. Modelling the travel times of residential travel data can help urban planners to predict future urban trends and traffic demand more accurately. This helps them to take traffic factors into full consideration when planning new roads, transport hubs or residential areas, making urban planning more scientific and rational. In this article, travelling time is used as a parameter for comparison, and the experimental results are shown in Fig. 7.

We discretise the 24 h of a day into 360 time points, and model the travel time of residents in 4-minute intervals. From Fig. 7, it can be clearly seen that our method has a much smaller probabilistic error in travelling time and on is closer to the real scenario. This is mainly due to the advantage of CNN in processing spatio-temporal data and its excellent performance in feature extraction. Comparing the urban traffic simulation methods based on UNIQ-SALT and RTDG models proposed in the literature, the comparison literature relies more on pre-defined or simpler features compared to the ability of CNN to extract features automatically, which limits the ability of the models to capture complex traffic patterns. Although the RTDG model can be adjusted to actual traffic data, it may not be as effective as LSTM in dealing with long-term time series dependence, because although the comparative literature proposes a method to approximate the actual data by iteratively calibrating the simulation results, this calibration process is not as flexible and efficient as the dynamic data compensation mechanism based on deep learning proposed in this article. Although the methods in the comparative literature show some advantages in model

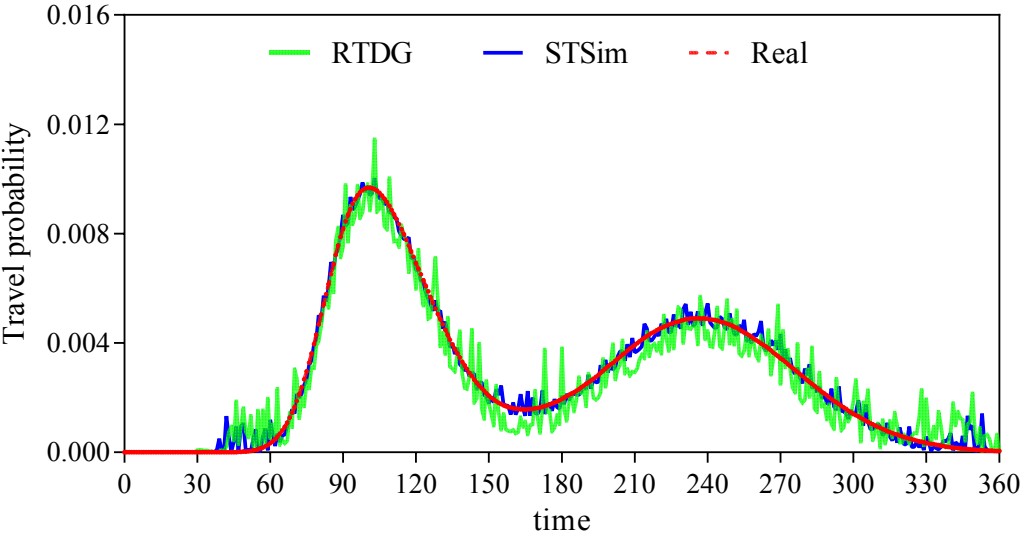

**Figure 7** Comparison of travel time probability distributions.

calibration, their computational efficiency and real-time performance are not as good as those based on deep learning in scenarios that require frequent updating of the model to adapt to new data. In addition, methods comparing the literature may perform well in specific scenarios, but their ability to generalise to different regions or different types of traffic networks may be limited, especially in the absence of sufficient real-time traffic data for calibration.

In addition to comparing with the baselines, in order to verify the role of different modules, ablation experiments are carried out, that is, three different model variants are obtained by replacing the corresponding modules in the proposed method, which are recorded as N-sp, N-tp and N-other, respectively. Specifically, N-sp deletes the spatial coding module, N-tp deletes the time interval coding module, and N-other deletes the external information coding module. Finally, Table 4 shows the metrics of different methods on three test data sets. The following results can be observed from this table. (1) The method based on deep neural network is better than other methods, the reason is that deep neural network can fit any function in principle, so using deep learning technology can obtain better results. (2) From the comparison of the results, it is not difficult to find that the spatial coding model is the most important module, followed by the important time interval coding module and external information coding module. In other words, the spatial coding module is the essential reason for the best effect of the proposed method.

## CONCLUSION

In this study, we experimentally validate the effectiveness of a deep learning approach combining CNN and LSTM in traffic data analysis. Our model can accurately extract and analyze the spatial and temporal features of traffic images and time series data to predict traffic flow patterns accurately. Compared to traditional statistical inference methods, our

**Table 4  Comparison of the results of different methods.**

| Methods | MAE | MAPE | MARE |
| --- | --- | --- | --- |
| N-sp | 110.42 | 23.67 | 22.05 |
| N-tp | 108.28 | 22.29 | 21.23 |
| N-other | 99.07 | 20.23 | 19.76 |
| RTDG | 137.52 | 29.04 | 27.42 |
| STSim | 93.37 | 18.06 | 19.17 |

model shows higher accuracy and flexibility, especially when dealing with complex and non-linear traffic patterns. Meanwhile, our approach reduces the need for large-scale data collection and significantly improves efficiency compared to survey methods. Overall, this study demonstrates the potential of deep learning in traffic data analysis, providing new perspectives for future traffic management and planning.

### Funding

This work was supported by the Prince Sattam bin Abdulaziz University through project number (PSAU/2023/01/26325). The funders had no role in study design, data collection and analysis, decision to publish, or preparation of the manuscript.

### Grant Disclosures

The following grant information was disclosed by the authors:
The Prince Sattam bin Abdulaziz University: PSAU/2023/01/26325.

### Competing Interests

The authors declare there are no competing interests.

### Author Contributions

- Adi Alhudhaif conceived and designed the experiments, performed the experiments, analyzed the data, performed the computation work, prepared figures and/or tables, authored or reviewed drafts of the article, and approved the final draft.
- Kemal Polat conceived and designed the experiments, performed the experiments, analyzed the data, performed the computation work, prepared figures and/or tables, authored or reviewed drafts of the article, and approved the final draft.

### Data Availability

The data is available at Zenodo: Kaggle. (2024). Travel details dataset [Data set]. Zenodo. https://doi.org/10.5281/zenodo.10907914.

The code is available at GitHub and Zenodo:
- https://github.com/kpolat14/codes--CNN-and-LSTM-/blob/main/CODE
- KEMAL, P. (2024). COMPUTER CODES- PYTHON CODES. Zenodo. https://doi.org/10.5281/zenodo.10907962.

## Supplemental Information

Supplemental information for this article can be found online at http://dx.doi.org/10.7717/
peerj-cs.2035#supplemental-information.

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
