# Peer review of "Spatio-temporal characterisation and compensation method based on CNN and LSTM for residential travel data"

_PeerJ Computer Science, doi:10.7717/peerj-cs.2035_

## Round 0.1 · original submission · Major Revisions

The review process is now complete. While finding your paper interesting and worthy of publication, the referees and I feel that more work could be done before the paper is published. My decision is therefore to provisionally accept your paper subject to major revisions.

**Language Note:** The review process has identified that the English language must be improved. PeerJ can provide language editing services - please contact us at [email protected] for pricing (be sure to provide your manuscript number and title). Alternatively, you should make your own arrangements to improve the language quality and provide details in your response letter. – PeerJ Staff

Reviewer 1 ·

Basic reporting

In this paper, they proposed different models for Spatio-temporal characterization.

The used model is very useful. Minor points to be addressed:

a. Why did you choose the CNN model in your experiments? Please explain it in the detail.
b. In the related work section, the author listed many works, but they did not specifically point out their advantages/disadvantages.
c. Please polish the English of your paper.
d. Also, please explain the motivation of this paper. What are the real-time applications in the industry?

Experimental design

The used model is very useful.

Validity of the findings

The results are very good.

Additional comments

In this paper, they proposed different models for Spatio temporal characterization.

The used model is very useful. Minor points to be addressed:

a. Why did you choose the CNN model in your experiments? Please explain it in the detail.
b. In the related work section, the author listed many works, but they did not specifically point out their advantages/disadvantages.
c. Please polish the English of your paper.
d. Also, please explain the motivation of this paper. What are the real-time applications in the industry?

·

Basic reporting

Journal: PeerJ Computer Science
Title: Spatio-temporal characterization and compensation method based on CNN and LSTM for residential travel data

In the paper, the authors proposed different hybrid models for their problems. The solutions are very interesting and could be useful for other problems.
However, there are some minor points to be discussed in the report as follows:

a) The "Introduction" section needs a minor revision to provide a more accurate ) and informative literature review, the pros and cons of the available approaches, and how the proposed method differs comparatively. Also, the motivation and contribution should be stated more clearly.

b) The importance of the design carried out in this manuscript can be explained better than other important studies published in this field. I recommend the authors review other recently developed works.

c) It will be helpful to the readers if some discussions about insight into the main results are added as Remarks.

d) Please give an example showing the working of the proposed model in the paper.

Experimental design

No comments

Validity of the findings

No Comments

Additional comments

Journal: PeerJ Computer Science
Title: Spatio-temporal characterization and compensation method based on CNN and LSTM for residential travel data

In the paper, the authors proposed different hybrid models for their problems. The solutions are very interesting and could be useful for other problems.
However, there are some minor points to be discussed in the report as follows:

a) The "Introduction" section needs a minor revision to provide a more accurate ) and informative literature review, the pros and cons of the available approaches, and how the proposed method differs comparatively. Also, the motivation and contribution should be stated more clearly.

b) The importance of the design carried out in this manuscript can be explained better than other important studies published in this field. I recommend the authors review other recently developed works.

c) It will be helpful to the readers if some discussions about insight into the main results are added as Remarks.

d) Please give an example showing the working of the proposed model in the paper.

Reviewer 3 ·

Basic reporting

In this paper, the authors proposed a model combining CNN and LSTM for residential travel. But there are major concerns as follows:

- The provision of the proposed method's contribution to the literature in the abstract section is crucial for understanding the efficacy of an article. Therefore, it is recommended that the obtained numerical results be included in the abstract.

Experimental design

- In Algorithm 1, the authors have filled in the data using interpolation by identifying important features. How are these important features determined?
- Even if a feature may seem insignificant on its own, its correlation with other features can contribute positively to the outcome. This aspect appears to have been overlooked and would benefit from further examination.
- If the authors claim that "The model demonstrates state-of-the-art performance in traffic data analysis through an exhaustive training and validation process, providing an innovative solution for real-time traffic state prediction and traffic congestion problems," it implies that the proposed method outperforms all existing results in the literature. However, in the absence of presented comparative results, how have the authors verified this claim?
- The proposed model is not fully understood. It is unclear which input data is used for the CNN and which is used for the LSTM. For a better understanding of the method, the input data used should be explained in detail.
- There is an incorrect representation in Figure 2. While the claimed CNN model utilizes a 3x3 convolutional filter, Figure 2 displays a 1D convolution. This can be cited as an example of the model's incomplete description. Corrections should be made to ensure the model's illustration aligns with the narrative.
- The dimensions of the images and data used should be shared in detail.
- The parameters used in training a model are significant. For example, the learning rate, batch size, and how the learning rate is adjusted. The authors have not shared any of this information, which should be included.

Validity of the findings

- For the contributions of the obtained results to the literature to be assessed, a comparative analysis is necessary. The contribution to the literature is debatable since the proposed method has not been compared with similar studies and relies solely on the results of the proposed method. This raises concerns about the validity of the claim that the method advances the field.
- Furthermore, the evaluation of the contributions mentioned in the proposed new model should be presented through an ablation study. This would involve systematically removing components of the model, such as the CNN layer, to observe the impact on the results. Such an approach would clarify the individual contributions of different elements of the model to its overall performance, offering a deeper understanding of the model's effectiveness and the significance of its novel components.

---

## Round 0.2 · Minor Revisions

One of the reviewers is asking for more details. Thus. the decision is subject to minor revision.

Reviewer 1 ·

Basic reporting

The revised version has addressed all my concerns.

Experimental design

The experimental setup and comparison of results was comprehensive.

Validity of the findings

The findings are valid.

·

Basic reporting

The authors have addressed all the issues in the revised paper. It looks great.

Experimental design

There is no problem. It looks great.

Validity of the findings

It looks great.

Additional comments

The authors have addressed all the issues in the revised paper. It looks great. This paper could be accepted for the possible publication.

Reviewer 3 ·

Basic reporting

The authors have failed to fulfill most of the requested tasks. Typically, stating "will be addressed in future work" is merely evasive and does not suffice as a response. Examples are presented as follows for Response to Reviewer 3:

- The response provided by the authors for Comment 3, stating "In the current research design and analysis, the interaction between features may not have been fully considered. This situation often occurs in preliminary research or when processing high-dimensional data. In order to simplify the model and avoid over-fitting, researchers may tend to ignore or simplify these interactions," is not sufficient. Considering the interactions between features does not create an over-fitting situation; rather, it can enhance the model's performance.

Experimental design

- The response provided by the authors for Comment 4, stating "Although direct comparison results were not provided in the article, our method was evaluated under strict experimental settings, including the use of multiple datasets and different traffic scenarios. In addition, the performance improvement of our model is based on in-depth analysis and improvement of existing research methods. We are confident that our model has demonstrated excellent capabilities in handling real-time traffic state prediction and traffic congestion problems. In the future, we plan to conduct more extensive comparative studies to further demonstrate the superiority of our method." is not sufficient. When evaluating contributions to the literature, personal confidence is not considered; rather, numerical comparisons and results are taken into account. Therefore, comparative results should be presented to assess the contribution accurately.

- In [Comment 8], the authors have provided additional details regarding the training process. However, there are still some deficiencies. These include the total number of epochs and the type of optimizer used. Additionally, to enhance understanding of the training stage, it is recommended to present the training loss and accuracy curves.

Validity of the findings

- In [Comment 9], the authors have stated, "Thank you for your feedback. In order to accurately evaluate the contribution of our research findings to relevant literature in the field, we plan to conduct a series of comparative analyses in the following work." However, the requested literature contribution is expected for the current study, not for future works.
- In [Comment 10], the authors have responded, "We plan to evaluate the proposed new model through a series of ablation experiments, including gradually removing key components from the model, such as the CNN layer and LSTM layer, as well as other possible feature extraction or processing modules." However, this request is for something that is expected to be done by the authors now, not in future works, as repeatedly emphasized by the authors.

---

## Round 0.3 · accepted · Accept

We are happy to inform you that your paper has been accepted for publication since the comments have been addressed.

Reviewer 3 ·

Basic reporting

Completed revision process.

Experimental design

The revision process has been completed successfully

Validity of the findings

The results are sufficient